# Applicability Study on Modified Argillaceous Slate as Subgrade Filling for High-Speed Railway

**Ping Hu** [1]**, Wei Guo** [2,*] **and Limin Wei** [2,*]

1   School of Engineering Management, Hunan University of Finance and Economics, Changsha 410205, China; huping@hufe.edu.cn
2   School of Civil Engineering, Central South University, Changsha 410018, China
*   Correspondence: Correspondence: guowei@csu.edu.cn (W.G.); lmwei@csu.edu.cn (L.W.)

**Abstract:** Modifying weathered argillaceous slate (WAS) to fill the subgrades of high-speed railways not only leads to obvious environmental benefits but also significantly saves natural resources. Previous studies have used several additives together with cement/quicklime. Using cement/quicklime as an additive would also improve the strength of weathered soft rock. Therefore, cement/quicklime can be used effectively as a sustainable solution. To illustrate the feasibility of cement/quicklime in WAS improvement, the mineral components and physical, mechanical and hydraulic characters for cement/quicklime-modified WAS were investigated via laboratory test. The dynamic behaviors were then analyzed by establishing a 3D vertical coupling dynamic model and performing dynamic triaxial tests. After mixing cement/quicklime, the following was proved: (1) The particles were coarsened, the content of kaolinite and amorphous phase increased, and the content of muscovite decreased, resulting in weakened hydrophilicity, reduced plasticity and enhanced water stability; (2) Mechanical and deformation characters were enhanced, reflecting an unconfined compression strength of quicklime/cement-modified soil greater than 0.5 MPa and elastic deformation for the subgrade of less than 0.1 mm; (3) For the quicklime-modified WAS, the dynamic strength was below the allowable value, and the sample disintegrated in water in 2 min, indicating it to be unsuitable for subgrade filling in HSR; (4) For cement-modified WAS, the water stability and dynamic strength were improved greatly, as well as the static strength. The sum of results highlights that 5% cement-modified WAS is recommended for filling subgrades of HSR for comprehensive factors.

**Keywords:** argillaceous slate; unconfined compression strength; dynamic response; water stability





## 1. Introduction

The main characteristics of soft rock are high clay mineral content, low strength, unstable performance, easy weathering, easy softening, easy disintegration, poor road performance and limited application in railways and highways. If it can be directly improved and used to fill the subgrade beds of high-speed railways, more than CNY 100 million of the project's investment can be saved. At the same time, spoiled soil can be reduced, the environment can be protected and the social and economic benefits are also very considerable. Ahmed Mancy Mosa [1] investigated the use of cement kiln dust to improve poor subgrade soil and found that improving poor subgrade soil leads to a reduction in the cost of construction by about USD 25.875 per square meter of pavements. Khiem Quang Tran [2] explored the effects of cornsilk fibers on the mechanical properties of cemented soil by conducting compaction, compression and splitting tension tests, and found that the largest improvements in compressive strength and splitting tensile are 177% and 88%. Similar studies have been reported while studying the effect of additives on other soils (as shown in Table 1). However, there is little research aimed at using WAS as the subgrade-filling material for HSR. Therefore, in order to minimize the cost of stabilization and the total energy associated with the entire process, further study is required before the recommended amounts of additives are used in the field of railway construction.

A/B-group filling (well-graded coarse-grained soil with less than 30% fine-grained soil) has been suggested to fill the bottom of the bed and the subgrade for HSR [3,4]. However, a large amount of C-group filling, namely weathered argillaceous slate (WAS), is exposed along the Wuhan–Guangzhou HSR, most of which is waste residue from railway cutting and belongs to soft rock. This is not suggested to fill the bottom of the bed and the subgrade directly in HSR [5–7]. If it can be modified to be used, it will protect the environment and save considerable investment.

In order to improve the mechanical properties of WAS, researchers have proposed the use of many kinds of stabilizer such as cement, lime, etc. [8–20], which are illustrated in Table 1. In these stabilizers, cement is widely used to improve soil's strength properties; however, cement is one of many reasons for the $CO_2$ emissions mentioned in [8]. Quicklime can permanently stabilize fine-grained soil, which chemically combines with water and improves soil workability and short-term strength. Lime modification works best in clay soils, as presented in [14]. Sand is also used to improve soft rock, through improving its graduation.

**Table 1.** Similar studies on adding additives to improve the properties of kinds of soil.

| Additives | Reference | Findings |
|---|---|---|
| Cement kiln dust | [9] | CKD can be an effective additive as it improves soil strength significantly |
| | [1] | (1) Adding CKD decreases the swelling ratio. <br> (2) Adding CKD reduces the cost of pavements by USD 25.875 per square meter. |
| Lime and cement | [10] | (1) Addition of lime/cement has noticeable impact on the geotechnical, textural and mineralogical characteristics of mountain soil. <br> (2) Satisfactory strength can be achieved with the addition of 5% additives to the soil mixture. |
| | [11] | (1) The mechanical properties of the modified filler are greatly improved. <br> (2) There is the best mixing ratio. |
| | [12] | (1) Mineral lattice structure was changed. <br> (2) Hydraulic and mechanical properties were improved. |
| | [13] | (1) The unconfined compressive strength of cement-modified red sandstone increases with the increase in cement content, curing age and compactness. <br> (2) There is an optimal lime content in the strength of lime modified red sandstone. |
| Lime | [14] | There is optimal lime content to improve the characters of red sandstone residual soil. |
| | [15] | All aspects of the improved filler have been better improved. |
| | [16] | (1) Cement has the greatest influence on the shear strength and compressive modulus of remolded red sandstone samples. <br> (2) Loess has a significant influence on the compaction effect of remolded red sandstone soil. |
| Water-resistant polymer cement base | [17] | The micro strain phase produced by polymer cement reinforced soil is three times higher than that of pure cement reinforced soil, showing good fatigue resistance. |
| Lime, cement and fly ash | [18] | (1) The modifier can greatly improve its bearing capacity and compressive strength, and its disintegration resistance and water stability are also improved. <br> (2) Among them, when the mixing ratio of cement-modified soil is 5%, the improvement effect is the best, and adding an appropriate amount of fly ash also improves the disintegration resistance. |

In recent years, the use of soft rock has been investigated through experiments, modeling and other methods. Some researchers studied the feasibility of modified soft rock as subgrade material by experiments [21,22]; Manasseh and Olufemi [15] and Garzon et al. [23] studied the effect of lime on some physical and geotechnical properties of Igumale shale and phyllites clays by indoor tests. The cement-modified method was studied by Xuesong

Mao [24] regarding the field performance of weathered phyllite as a subgrade material for highways, and was also investigated by Al-Rawas et al. regarding expansive soil [25]. Moreover, Guo studied the characterization of the geomechanical and water-vapor-absorption properties of deep soft rocks [26]. Modeling is another means of studying the feasibility of modified soft rock as subgrade material [27]. D. Xiao investigated the influence of cement-fly ash-gravel pile-supported approach embankments on abutment piles in soft ground [28]. Zheng et al. (2013) performed a numerical analysis to study the mechanical properties of soft soil beneath approach embankments [29].

From the analysis above, it is clear that most previous studies were about the performance of modified soft rock for highways. There is little research aimed at using WAS as the subgrade-filling material for HSR. Moreover, the post-construction settlement can not exceed 15 mm due to the high requirements of deformation control of the HSR subgrade; whether modified WAS can be used in the bottom bed or subgrade in HSR requires further research.

Based on the experiences of Japan, Europe and China, soft rock can be used as subgrade filling for HSR through modification, and must meet some requirements [30,31].

First, the allowance of elastic deformation is 1.5 mm in Japanese HSR, compared to 2.0 mm in China's HSR and 1.0 mm in European HSR [32–35].

The allowable dynamic strength of the modified soil should then meet the following requirements [3,4]:

$$\sigma_{bcu} \geq \frac{\beta \cdot \sigma_{zl} \cdot K_h}{\eta_g \cdot R_{cr}} \tag{1}$$

where $\sigma_{bcu}$ is the allowable dynamic strength, kPa; $\beta$ is the dynamic stress fluctuation coefficient, which is suggested to be 1.2, kPa; $\sigma_{zl}$ is the dynamic stress induced by train load, kPa; $\eta_g$ is the strength attenuation coefficient of dry and wet cyclic, which is suggested to be 0.7–0.95; $K_h$ *is* the compaction coefficient (heavy compaction system); and $R_{cr}$ is the dynamic-static ratio, which is recommended to be 0.45.

Additionally, according to the Research Institute of Railway Comprehensive Technology of Japan, when the saturated unconfined compressive strength (UCS) of the modified soil is greater than 500 kPa, both the water stability and dynamic stability can meet the filling requirements of the subgrade for HSR.

Literature [10] demonstrated the engineering performance of the modified mixture, which was evaluated based on elastic deformation, static strength, dynamic strength and water stability. This evaluation confirms the feasibility of using modified WAS as a bottom-of-bed and subgrade material for HSR. Moreover, cement and quicklime are the best additive for soft rock to improve the UCS. Therefore, the present investigation described in this paper studies the use of cement and quicklime to modify the performance of WAS as a bottom-of-bed and subgrade filling for HSR.

## 2. Materials and Test Program

### 2.1. Characteristics of WAS

The WAS samples were collected from test section of DK1412 + 195~DK1412 + 465 along Wuhan–Guangzhou HSR (Figure 1a,b). It was easy to hand-knead into powder, and slightly slippery, with foliation structure and fine mineral particles. The chemical and mineral composition of WAS (Tables 2 and 3) was determined by performing Energy Dispersive X-ray Spectro Analysis (EDS), The results illustrate that WAS is mildly acidic, and $SiO_2$, $Al_2O_3$ and $Fe_2O_3$ are the main chemical compositions, accounting for 75.0–89.0%; the amount of muscovite and quartz is up to 46.25% and 24.97%, respectively, demonstrating that WAS is easily weathered and softened.

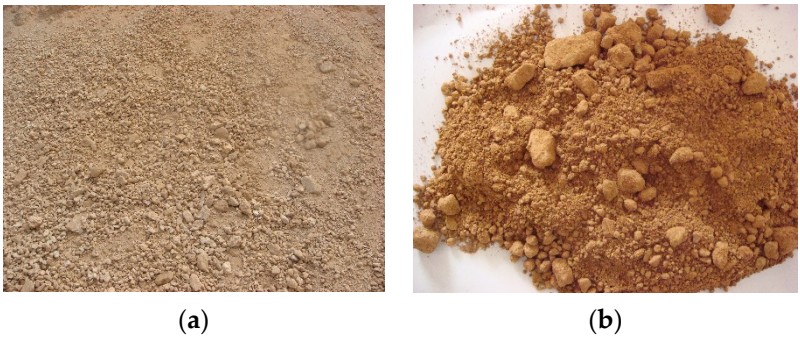

|  (**a**)  |  (**b**)  |

**Figure 1.** Weathered argillaceous slate samples. (**a**) Soil sample in the field; (**b**) Soil sample in laboratory.

**Table 2.** Quantitative analysis of chemical composition of WAS by EDS (%).

| Constituents | $SiO_2$ | $TiO_2$ | $Al_2O_3$ | $Fe_2O_3$ | MgO | CaO | $K_2O$ | $Na_2O$ |
|---|---|---|---|---|---|---|---|---|
| Percentage(%) | 56.20 | 1.50 | 21.40 | 11.58 | 1.40 | 0.48 | 3.53 | 0.80 |

**Table 3.** Average mass percent of minerals (%).

| Phase-Name | Clinochlore | Muscovite | Kaolinite | Albite | Quartz | Rutile |
|---|---|---|---|---|---|---|
| Average mass percent(%) | 24.31 | 46.25 | 3.65 | 0.63 | 24.97 | 0.20 |

The physical properties of the WAS, namely specific gravity, optimum water content, maximum dry density and liquid-plastic limits, were determined as per Code for Soil Test of Railway Engineering (TB10102-2010) [33], and the results are tabulated in Table 4. The grain-size analysis suggests that grading indexes of WAS are Cc = 0.61 and Cu = 42 (Figure 2). Thus, the WAS was categorized as C-group filling [33].

**Table 4.** Physical properties of WAS.

| Natural Water Content $w$ (%) | Specific Gravity ($g/cm^3$) | OMC (%) | MDD ($g/cm^3$) | Plastic Limit $w_p$ (%) | Liquid Limit $w_L$ (%) | Plasticity Index Ip |
|---|---|---|---|---|---|---|
| 10.8~13.5 | 2.72 | 12.4 | 1.844 | 22.4 | 33.2 | 10.4 |

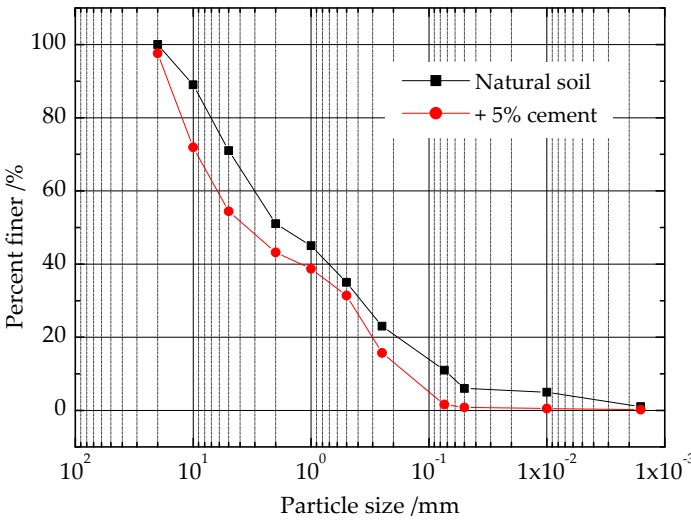

**Figure 2.** Comparison of gradation curves of natural and modified WAS.

## 2.2. Design of Test Scheme

Tests were carried out on WAS with addition of quicklime (5%, 6% and 7%) and cement (3%, 5% and 7%) to assess the changes in engineering characters and provide parameters in dynamic simulation. Tables 5 and 6 present the chemical composition of quicklime and cement.

**Table 5.** Quicklime detection indicators used in tests.

| Constituents | CMC (%) | MgO (%) | S (%) | Si (%) | URC | $CO_2$ Content (%) | Pulp Yield (%) | Grade |
|---|---|---|---|---|---|---|---|---|
| Value | 92.80 | 0.96 | 0.006 | 0.058 | 4 | 4 | 4.8 | excellent |

Note: "CMC" means CaO + MgO content; "URC" means undigested residue content (%) (residue of 5 mm round-hole sieve).

**Table 6.** Main performance indicators of 32.5 ordinary Portland cement.

| Constituents | $SO_3$ | MgO | LOI | R80 RHS | Specific Surface Area | Setting Time | | Strength | | | |
|---|---|---|---|---|---|---|---|---|---|---|---|
| | | | | | | Initial Setting | Final Setting | Flexural | | Compression | |
| | (%) | (%) | (%) | (%) | (m$^2$/kg) | (Hour) | (Hour) | 3 Days | 28 Days | 3 Days | 28 Days |
| Value | 3.44 | 2.58 | 4.37 | 3.5 | 328.5 | 3:17 | 6:21 | 4.5 | 6.2 | 20.3 | 34.4 |

Note: "LOI" means loss on ignition; "R80 RHS" means residue of 80 μm round-hole sieve.

The experimental program in this study was performed according to TB10102-2010 [33] and ASTM D559-44 [36,37], as shown in Table 7.

**Table 7.** Details of experimental program.

| Additives | Sample (Size, Shape) | Test Type | Test Machine | Standard |
|---|---|---|---|---|
| +3%, 5% cement + 6% quicklime | | EDS | SEI (3.0 nm) BEI | |
| +5%, 6%, 7% quicklime + 3%, 5%, 7% cement | 20 mm sieve | Z2 Compaction test | DJ30-5 | TB10102-2010 |
| +5%, 6%, 7% quicklime + 3%, 5%, 7% cement | Φ50 mm × 50 mm 10 mm sieve | UCS test | YYW-2 | TB10102-2010 |
| +5%, 6%, 7% quicklime + 3%, 5%, 7% cement | 2 mm sieve at 0.92 compactness | shearing test | SZ304 | TB10102-2010 |
| +3%, 5%, 7% cement Quicklime (collapsed) | Φ50 mm × 50 mm 10 mm sieve | dry-wet test | YYW-2, basin | ASTMD 559-44 |
| Natural WAS | Natural air-dried soil | specific gravity | Pycnometer | |
| +4%, 6%, 8% quicklime + 4%, 5%, 6% cement | 0.5 mm sieve | liquid-plastic limits | GYS-2 | TB10102-2010 |
| natural +5% cement | Natural air-dried soil | Particle size | seives | TB10102-2010 |

Literature [10] demonstrated that there is an optimum content for cement or quicklime to improve the USC of the weathered soft rock, which is 6% for quicklime and 5% for cement. In order to investigate the best content for use as additive, a wide range of cement content (3%, 5%, 7%), and quicklime content (5%, 6%, 7%) were considered in the present work.

First, EDS was carried out to understand the mineral component of natural and modified WAS. Compaction tests were then carried out to measure the maximum dry density (MDD) and the optimum moisture content (OMC), which can be used to control the compaction quality of the subgrade filling. Next, unconfined compression strength (UCS) test and shearing test were carried out to obtain the mechanical characteristics of the natural and modified WAS. Finally, in order to obtain the water stability of WAS, dry-wet tests were conducted on the modified WAS.

### 3. Experimental Evaluation of Cement/Lime-Modified Soil

*3.1. Results*

3.1.1. Analysis of Main Mineral Components for Natural and Modified Soil

Mineral composition of the modified soil after 7 days of curing was carried out by performing EDS. These analyses help in understanding and appreciating the increase in geotechnical properties of lime/cement-modified soil at a microscopic level. The mineral composition of the soil modified by 3% and 5% cement and 6% quicklime are presented in Table 8.

**Table 8.** Analysis of mineral components of modified Soil.

| Phase-Name \\ Samples | Quartz (%) | Muscovite (%) | Illite (%) | Feldspar (%) | Kaolinite (%) | Siderite (%) | Calcite (%) |
|---|---|---|---|---|---|---|---|
| Natural soil | 47.54 | 36.42 | 4.47 | 1.57 | 3.59 | 5.68 | 2.87 |
| +3% cement | 49.17 | 33.82 | 3.23 | 1.70 | 3.78 | 4.27 | 3.65 |
| +5% cement | 55.90 | 26.85 | 2.82 | 1.93 | 3.87 | 3.54 | 3.79 |
| +6% quicklime | 52.37 | 28.47 | 3.46 | 1.82 | 3.62 | 3.26 | 3.53 |

The muscovite content is recorded to decrease from 36.42% to 26.85%; simultaneously, illite decreases lightly. It weakens the hydrophilicity of the soil, enhances water stability and reduces the plasticity of the soil.

The quartz content is noted to increase from 47.54% to 55.90%; simultaneously the contents of kaolinite, feldspar and calcite increase slightly, showing that the total content of clastic minerals—particularly the amorphous phase that is mostly formed by $SiO_2$—increases sharply. Similar observations were reported while studying the effect of lime on expansive clays [38–40]. This may be attributed as the result of crystallization, a part of amorphous phase generated by ion exchange, condensation and pozzolanic reaction after mixing additive; the other part is from the additive (quicklime, cement) itself which is generated by calcination. It not only increases the size of soil particles, but also makes the soil form a new slab structure, thus enhancing the strength and stiffness of the soil.

3.1.2. Analysis of Basic Physical Index for Natural and Modified Soil

Liquid-plastic limit tests of natural and modified soil were carried out according to [33]. The specimen was sieved through a 0.5 mm sieve, and was then tested after setting for 24 h. The water content corresponding to the sinking depth of 10 mm and 2 mm is the LL and PL. Thereafter, the plasticity index (PI) was calculated by taking the difference between LL and PL, and those indexes with varying percentages of quicklime/cement are illustrated in Table 9.

**Table 9.** Test results of combined determination of liquid-plastic limit.

| Physical Index | Mixture | Liquid Limit LL (%) | Plastic Limit PL (%) | Plasticity Index PI (%) |
|---|---|---|---|---|
| Curing 1 day | +4% quicklime | 23.26 | 23.26 | 9.97 |
| | +6% quicklime | 24.11 | 24.11 | 8.47 |
| | +8% quicklime | 23.45 | 23.45 | 8.40 |
| Curing 3 days | +4% cement | 24.07 | 24.07 | 10.92 |
| | +5% cement | 25.23 | 25.23 | 7.47 |
| | +6% cement | 24.16 | 24.16 | 7.07 |

Table 3 indicates that the WAS belongs to low-liquid-limit soil and silty soil with weak stability. After being modified, LL and PI decrease considerably with the addition of quicklime/cement. PL increased from 22.4% to 24.11% up to 2% addition of quicklime,

while further addition of quicklime caused a slight reduction. The same changing rule was applied for cement-improved WAS. Hence, the resulting PI shows considerable reduction, reaching up to 10%. The amount of admixture has an effect on the plasticity of soil; the comparative study indicates that the reduction in the PI of mountain soil is relatively higher in the case of cement-treated soil when compared to lime-treated soil. The results have good agreements with other studies on soil–lime and soil–cement mixtures [41–44]. This results in a reduction in plasticity, which thereby enhances 'workability'. The increase in the consistency limits of soils upon the addition of lime can be attributed to the increase in affinity of clay particles to water. However, expansive clays are rather inert to the action of hydroxide.

### 3.1.3. Particle-Size Distribution for Natural and Modified Soil

Figure 2 shows the grain-size distribution curves of natural and modified soil WAS (+5% cement). The grain distribution curve of WAS moves to the left after being modified, showing a decrease in clay content. This may be attributed to: (1) the hydration products of cement producing a cementation effect on soil particles and agglomerating soil particles; (2) the PH value increasing to a certain extent, stimulating the activities of $SiO_2$ and $Al_2O_3$ in clay minerals, and reacting with $Ca^{2+}$ in solution to form calcium silicate and calcium aluminate series minerals. These new minerals have the same characteristics. With cementing power, they encapsulate the surface of clay particles and coagulate small clay particles into large particles together with the hydration products of cement. Similar observations have been reported while studying the effect of lime on expansive clays [25,45].

### 3.1.4. Compaction Characteristics

The variation in MDD and OMC at different quicklime/cement contents is illustrated in Figure 3. The samples were sieved through a 20 mm sieve and the tests were evaluated by Z2 compaction [33].

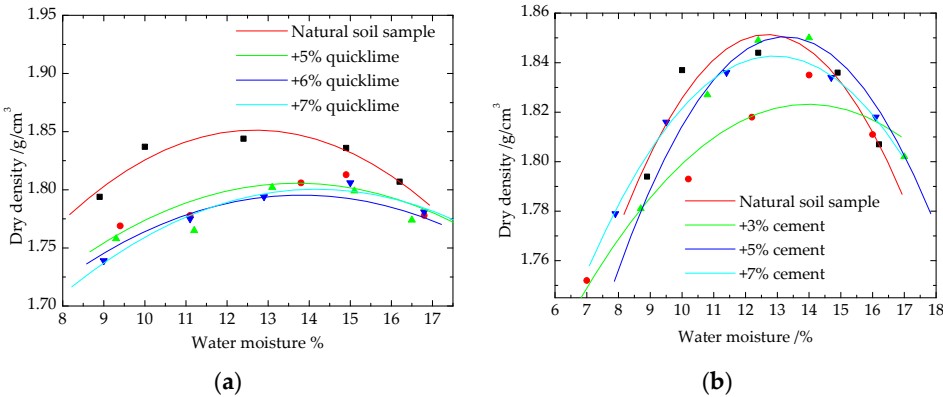

**Figure 3.** Compaction curves of modified soil. (**a**) Quicklime-modified soil; (**b**) Cement-improved soil.

The increase in OMC for quicklime-modified WAS is recorded from 12.4% to 15%, while for cement-treated soil it is noted to have increased up to 14%. The MDD decreased from 1.844 g/cm³ to 1.804 g/cm³ for lime-modified WAS, while in the case of cement-modified WAS, it decreased to 1.835 g/cm³. With the increase in the mixing ratio of quicklime and cement, the compaction curve tends to be flatter and the limit of moisture content in the error range increases gradually. An increase in the specific surface area of the mixture and the cation exchange reaction may attribute to the increases of the OMC. Additionally, the low density of cement/quicklime and the chemical reactions may lead to the decrease in MDD of the cement/lime-modified WAS. Similar observations have been reported by other researchers [46], who had utilized lime, cement or their admixture to enhance the geotechnical properties of various type of soils [47–50].

### 3.1.5. Unconfined Compression Strength Characteristics

UCS is a vital mechanical parameter used in soil stabilization and construction of subgrade. UCS tests were conducted on the Φ50 mm × 50 mm quicklime/cement-modified cylinder samples with optimum moisture. The samples were sieved through a 10 mm sieve.

Figure 4 shows the relation between UCS and compactness of specimens cured for 7 days. The results suggest that UCS increases with both compactness and quicklime/cement content. The compactness enhances the UCS sharply, while the addition of cement is 5% and the addition of quicklime is 6%. Similar comparative analysis between the effect of lime and cement suggests that the UCS of soil–cement mixture is higher than the soil–lime mixture [51,52]. This has been attributed to the increase in the concentration of $Si^{+4}$, $Al^{+3}$ and $K^+$ ions on the addition of cement, in contrast to the addition of lime.

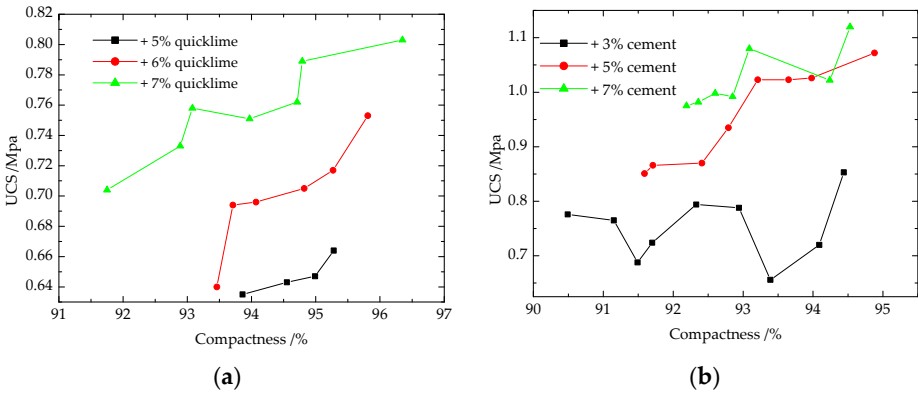

|      |      |
| :--: | :--: |
| (**a**) | (**b**) |

**Figure 4.** Relation between UCS and compactness of modified WAS after 7 days of curing. (**a**) Quicklime-modified soil; (**b**) Cement-improved soil.

Figure 5 compared the UCS of air-dried WAS and immerged WAS. For the specimen modified with quicklime, it collapsed easily if the curing time was less than 28 days. Even if the curing time reaches 27 days, the UCS of the specimen immerged in water is almost one-third of that which was not immerged in the water, demonstrating a very weak water stability. For the specimen modified by cement, the value of the increase in UCS due to increased curing time is greater than the value of the decrease in UCS due to immerging in water, indicating that the cement-improved soil with enough maintenance time has good water stability. Additionally, whatever the mixture is, the UCS after 7 days' maintenance is greater than 0.5 MPa, showing that the quicklime/cement-modified WAS can meet the static strength requirement for high-speed subgrade filling.

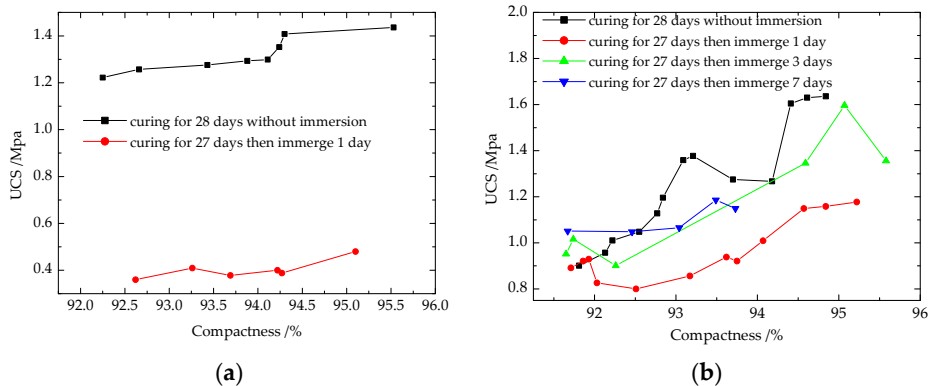

|      |      |
| :--: | :--: |
| (**a**) | (**b**) |

**Figure 5.** The relationship between UCS (saturated and unsaturated) and compactness. (**a**) +6% quicklime-modified soil. (**b**) +5% cement-improved soil.

### 3.1.6. Shearing Characteristics

The shear strength of soil is one of the important mechanical indexes of soil and a key factor to determine the stability of subgrades and geotechnical structures. The specimen was sieved through a 2 mm sieve, and with OMC at 0.92 compactness, was conducted by direct shear test. Table 10 and Figure 6 show the results.

**Table 10.** Shear test parameters and fitting correlation coefficients of modified WAS.

| Soil Sample | Mixture Percent (%) | Internal Friction Angle φ(°) | Cohesion c (kPa) | Fitting Correlation Coefficient R |
|---|---|---|---|---|
| Natural sample | - | 27.6 | 59.0 | 0.995 |
| quicklime | 5 | 28.2 | 62.1 | 0.99 |
|  | 6 | 29.6 | 76.5 | 0.992 |
|  | 7 | 29.2 | 67.3 | 0.994 |
| cement | 3 | 29.0 | 68.5 | 0.998 |
|  | 5 | 30.6 | 79.9 | 0.998 |
|  | 7 | 31.8 | 100.8 | 0.999 |

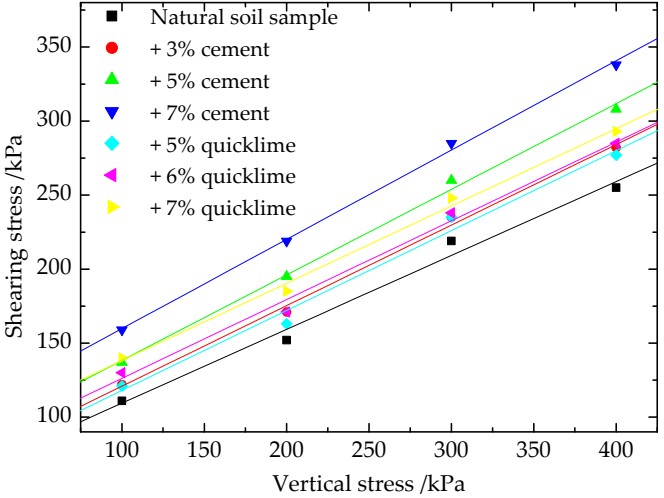

**Figure 6.** σ-τ curve of shear tests for natural and modified WAS.

For the quicklime-modified WAS, the cohesion and internal friction angle increases and reaches the maximum value when the additive content is 6%. Further addition of additives adversely affects the cohesion and internal friction angle. For the cement-modified soil, the cohesion and internal friction angle increases with the increase in cement content. This can be attributed to the chemical reactions (pozzolanic and cation exchange) which occur between soil and additives. Additionally, under the same conditions, the C and φ values of cement-improved soil are larger than those of quicklime-improved soil. The maximum cohesion and internal friction angle in the case of 6% quicklime-modified soil is therefore 76.5 kPa and 29.6°, while the maximum cohesion and internal friction angle in case of 7% quicklime-modified soil is therefore 100.8 kPa and 31.8°. One important factor is that cement is a quick-setting material and its strength reaches 60–70% of the total strength after 7 days, which makes the internal friction angle and cohesion C of the modified soil increase rapidly. Meanwhile, quicklime is a retarding material whose strength increases slowly after 7 days, and the total strength generally takes 5–10 years to reach a stable value; therefore, the internal friction angle and cohesion C of the quicklime-improved soil increase slowly. Similar observations have been reported while studying the effect of lime on other soils [2,53].

### 3.1.7. Dry-Wet Cycle Characteristics

The dry-wet cycling caused by the variability of environmental conditions is an important factor that leads to the degradation of the mechanical properties of railway subgrade filling, and consequently leads to subgrade failure in practical engineering. The ability of modified soil to withstand the dry-wet cycle reflects its ability to withstand water damage in natural environment. This is one of the most important indexes of modified soil's durability. According to ASTMD 559-44 and ASHOT 136-45 "Standard Method of Compacted Cement-Soil Dry-Wet Test", the weight loss of modified soil caused by the dry-wet cycle is determined. The standard curing specimen is weighed after being immersed in water for five hours at room temperature, and then is dried in an oven at 71 °C for 42 h. The floating particles on the surface of the specimen are brushed off by a steel-wire brush, weighed again; this is then repeated 12 times. The weight-loss percentage of the modified soil is used to judge its quality and stability. According to this test method, the variation of strength and saturated mass loss of different dry-wet cycles for cement-improved specimens are shown in Figures 7 and 8, while the quicklime-modified WAS quickly collapsed (Figure 9).

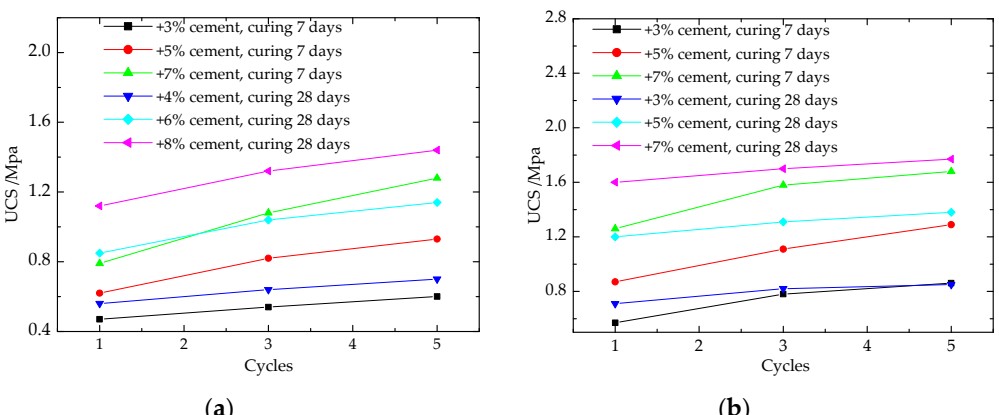

(a)  (b)

**Figure 7.** The relationship between UCS and the dry-wet cycle times. (**a**) 0.92 of compactness. (**b**) 0.95 of compactness.

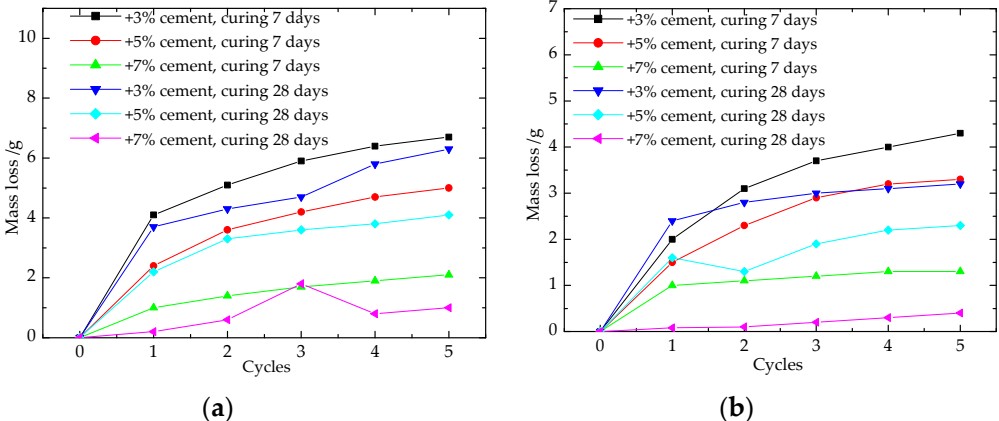

(a)  (b)

**Figure 8.** The relationship between mass loss in saturated state and cycle times. (**a**) Compactness $k_h = 0.92$; (**b**) Compactness $k_h = 0.95$.

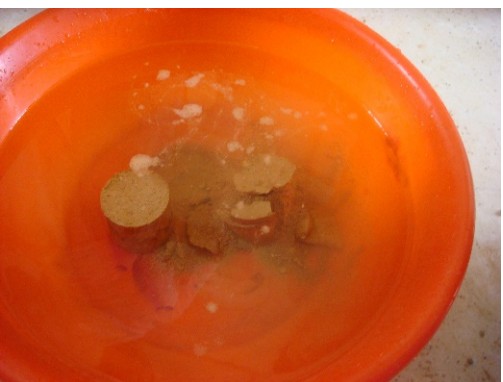

**Figure 9.** Soaking disintegration of modified-soil WAS (left: cement, right: quicklime).

From Figure 7, it can be seen that the USC of cement-improved soil increases with the increase in dry-wet cycles. This phenomenon may be attributed to the greater influence of curing age on USC than that of cement admixture. In addition, with the increase in cycles, the free water in the sample decreases and the bound water increases gradually until they reach a certain dynamic equilibrium, and this equilibrium state changes continuously with the change in external environmental factors (such as temperature) to form a new dynamic equilibrium. In this way, the bound water gradually replaces part of the free water to fill the pore, forming a stable structure with the skeleton structure of the soil, which increases the strength of the sample.

From Figure 8, it can be seen that after three dry-wet cycles, the quality of the WAS tends to be stable, and the higher the cement content, the better the water stability of the WAS is and the lower the mass loss of the dry-wet cycles is. After five dry-wet cycles, the mass loss ranges of the WAS with 3%, 5% and 7% cement are about 4–7 g, 2–5 g and 1–3 g, respectively, accounting for 3.2%, 2.3% and 0.8% of the total mass.

When the cement-modified sample is being saturated in water, the water immersion causes small bubbles to emerge on the surface and surround the sample. Initially, the water bubbles up for nearly 10 min. Thereafter, bubbling slows down until the water in the soil tends to balance with the water in the soil, i.e., until the sample is saturated, and the sample remains intact. This therefore means that the water stability of the cement-improved soil is good.

The sample of quicklime-modified soil after 14 days' curing disintegrated in water is shown on the right side of Figure 9. Owing to the presence of air in the WAS and the invasion of water, small bubbles appeared on the surface and surrounded the modified soil. Initially, bubbles popped out quickly. After 7 min, the WAS was softened, cracks appeared in the soil and the surrounding particles continued to fall off until the WAS collapsed. This demonstrates that the water stability of the quicklime-improved soil is bad.

### 3.2. Discussions

Mixing quicklime/cement makes the content of muscovite decrease considerably, while making the content of kaolinite and clastic minerals, especially amorphous phase which is composed of $SiO_2$, increase greatly, and coarsening the particles of WAS. Therefore, the physical, mechanical, water stability and dynamic performances of the modified WAS are modified by adding quicklime or cement.

The water stability of WAS was modified by mixing cement or quicklime. For cement-modified soil, even if the WAS experienced five dry-wet cycles, the mass-loss ranges of the WAS with 3%, 5% and 7% cement are only about 4–7 g, 2–5 g and 1–3 g, respectively, accounting for 3.2%, 2.3% and 0.8% of the total mass. The UCS even increases with days of curing after dry-wet cycles, showing good water stability. For the quicklime-modified WAS, it disintegrates fast after immerging in water, and the UCS of the specimen cured at 28 days is less than 0.5 MPa, showing weak water stability.

## 4. Dynamic Simulation Analysis of Soft-Rock-Modified Soil as Subgrade

At present, there are three kinds of transition section models based on numerical analysis: one is the train-track coupling model [4], the other is the track-subgrade coupling model [40,41,46], and the third is the vehicle-track-subgrade model [42]. In this study, a track-subgrade coupling model was established; the effect of the train was modeled by a moving train load.

### 4.1. Dynamic Model Establishment

4.1.1. Dynamic Fundamentals

The ballastless HSR track-subgrade three-dimensional space coupling system consists of several subsystems, including the rail, fastener, the pad under the rail, track plate, CA mortar, integral concrete track bed, bed layer and subgrade layer. The total energy is a sum function as follows:

$$W(t) = W(t)_{\text{int}} + W(t)_{ext} + W(t)_{\text{int}} + W(t)_{kin} + W(t)_{dam} + W(t)_{coup} \tag{2}$$

where $W(t)_{\text{int}}$ is the strain energy of all the subsystem; $W(t)_{ext}$ is the work of various external forces such as wheel forces; $W(t)_{kin}$ is the kinetic energy of all the sub-structure; $W(t)_{dam}$ is the work of damping force including structural damping and material damping; $W(t)_{coup}$ stands for the work of the binding which is the increasing deformation energy created by the system of each substructural unit's displacement coupling constraints.

When the structure moves from one equilibrium position to an adjacent position that is infinitely close to that equilibrium position, the total energy $W(t)$ keeps constant. That is to say, the energy variation of the system has a stationary value:

$$\delta W(t) = 0 \tag{3}$$

Then, taking variants of Formula (2):

$$\delta W(t) = \delta W(t)_{\text{int}} + \delta W(t)_{ext} + \delta Wt)_{\text{int}} + \delta W(t)_{kin} + \delta W(t)_{dam} + \delta W(t)_{coup} = 0 \tag{4}$$

Therein, $W(t)_{coup}$ was established by the Lagrange augmentation method, which is a combination of the Lagrange multiplier and penalty function method. The system equation stiffness matrix turns to be non-definite after introducing the excess unknown variables. At the same time, for the given penalty $\alpha$ and Lagrange multiplier $\lambda$, the modified energy functional $W^*$ is obtained as follows:

$$W^*(u, \alpha, \lambda) = W(u) + \lambda^T g(u) + \frac{1}{2}\alpha g^T(u)g(u) \tag{5}$$

where $g(u) = 0$ is the constraint equation; $W(u)$ is the energy $W_{coup}$ divided by formula (1); and $u$ is the displacement matrix.

The Lagrange multiplier was used to determine coupling constraint equations. The energy of each subsystem in the model was determined and the coupling energy of each different element was integrated into the total energy matrix. The whole coupling system energy of a subgrade rail system is worked out at time $t$. Finally, the dynamic matrix equation of each substructure of a ballastless subgrade-rail system can be obtained using the stationary energy principle. Thus, the total energy of the system is a sum of the energy of the above subsystems and using different elements of various characteristics will discretize it spaciously. The equation was resolved using the Newton–Raphson method [27,31].

4.1.2. Model Establishment and Verification

The calculation length along the line is taken as 63 m; the depth of the subgrade is taken as 5 m. Considering the symmetry of the doublet structure for HSR, only half of the railway line was used in the calculations. The XZ plane is the symmetry plane seen in Figure 10.

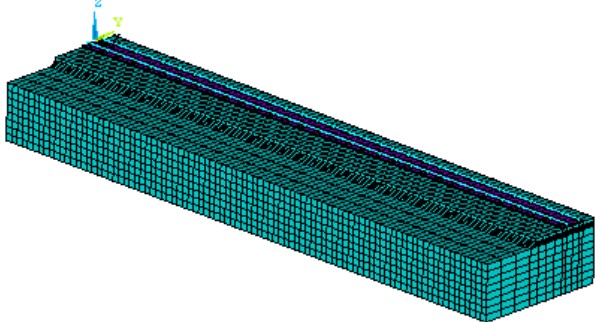

**Figure 10.** Numerical Model.

The whole rail-track coupling system was established by finite element software ANSYS and the corresponding APDL language [44,45]. The subsystems were modeled by different elements, while coupling surfaces of the separate materials were connected to each other without relative displacements. Horizontal fixing was constrained in the symmetrical plane, and the viscoelastic artificial boundary was used in the model at the cutoff of an infinite continuous medium. In addition, Rayleigh damping and a viscoelastic constitutive model were used in this paper.

Parameters: Fasteners and a sub-track rubber cushion system were considered as a spring-damp structure with an initial length of 38 m. Adopted total equivalent stiffness was $4.38 \times 10^7$ N/m, and the damp parameter was $4.5 \times 10^5$ N · s/m, with a CA mortar elastic modulus of $1.25 \times 10^9$ N/m³ and damping of $8.3 \times 10^4$ N · s/m². Other general calculation parameters are listed in Table 11. The parameters with * marked in the table are estimated from tests mentioned in Section 3. The additional parameters are generic values widely adopted in the literature.

**Table 11.** Calculation model parameter of tunnel-culvert-culvert transition zone.

| Parameters | $E$ (GPa) | $v$ | $\rho$ | $\Phi$ (°) | $C$ (kPa) |
|---|---|---|---|---|---|
| Rail | 210 | 0.3 | 78 | | |
| Track plate | 34.0 | 0.1667 | 30 | - | - |
| Concrete Supporting | 25.5 | 0.20 | 24 | | |
| C15 concrete | 22.0 | 0.25 | 24 | - | - |
| Graded gravel +5%cement | 1.78 | 0.30 | 24 | 45.8 | 1326 |
| Graded gravel | 1.58 | 0.24 | 24 | 39.5 | 160 |
| A/B-group filling | 0.53 | 0.26 | 2.35 | 41.8 | 200 |
| +3% cement | 0.28 | 0.26 | 1.61 * | 29.0 * | 68.5 * |
| +5% cement | 0.32 | 0.26 | 1.62 * | 30.6 * | 79.9 * |
| +7% cement | 0.36 | 0.26 | 1.65 * | 31.8 * | 100.8 * |
| +5% quicklime | 0.22 | 0.26 | 1.58 * | 28.2 * | 62.1 * |
| +6% quicklime | 0.24 | 0.26 | 1.59 * | 29.6 * | 76.5 * |
| +7% quicklime | 0.26 | 0.26 | 1.57 * | 29.2 * | 67.3 * |

Train load: Since the model was used to study the dynamic performance of track and subgrade, the trainload was simplified by a moving wheel-rail force. As shown in Figure 11, At any time $t$, when vehicle moves in a constant speed the wheel loads are:

$$f(t) = p\delta(x - vt) \tag{6}$$

where $p$ is wheel load; $x$ is the wheel position at time $t$; $\delta$ is function of $\delta$, viz., $\delta(x_i = vt_i) = 1$ and $\delta(x_i \neq vt) = 0$, in which $x_i$ is rail ditch position of the track plate, $x_i = x_0 + il_s(i = 1, 2, \dots, N)$; $x_0$ is the initial referring point; and $l_s$ is the space between rail ditch. Taking the train (CRH2-300) speed $v = 350$ km/h, two power carriages at the head and tail and six carriages

in the middle of the train were included. For the power carriage, the coach distance is 11.46 m, the wheelbase is 2.5 m, the vehicle length is 25 m, the axle weight is 14 t and the wheel load is 70 kN. For the trailer, the wheelbase is 2.5 m, vehicle length is 25 m, axle load is 14 t and wheel load is 70 kN.

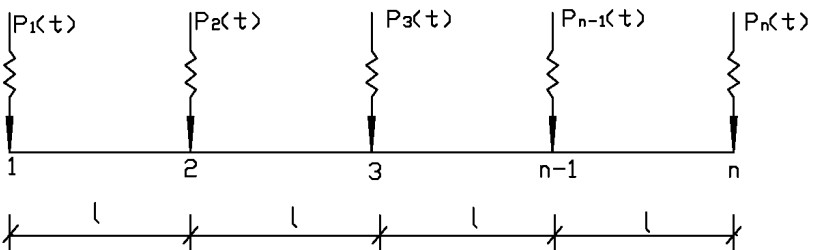

**Figure 11.** Diagram of dynamic load application.

Track and subgrade: The rail was made of No. 60 rail in Japan and the cross-section was an "I"-shaped beam; the space between rails was 1435 mm. The rail was simulated by the Timoshenko beam element, while the track plate was modeled via plate element. The fasteners between the rail and track plate and the pad structure system under the rail were simulated using spring-damping element, while CA mortar layer was established by shell element together with spring-damping element. The bed layer and subgrade layer were established by three-dimensional solid elements.

4.1.3. Verification

The proposed method is validated by in situ results measured from Wuhan–Guangzhou HSR. WAS +5% cement was used for subgrade in simulation analysis, while A/B-group filling was used to fill the subgrade in test section. The dynamic displacement rules that changed along the depth were compared between the results of the simulation and test results (Figure 12), proving that the finite element models established in this paper are essentially correct.

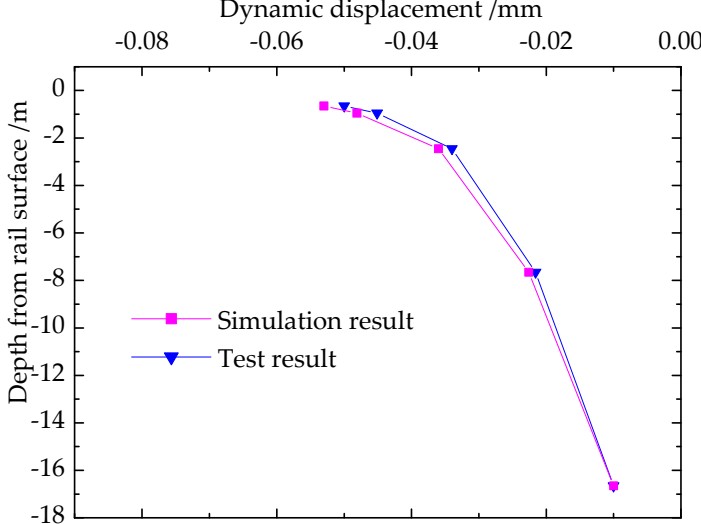

**Figure 12.** Comparison of dynamic displacement between simulation and test results.

*4.2. Results of Dynamic Response Analysis*

4.2.1. Dynamic Response Analysis

To study the dynamic behaviors of subgrade filled with modified WAS, A/B-group subgrade-filling material was substituted by either WAS + 6% quicklime or WAS + ce-

ment (3%, 5%, 7%) in the numerical simulation. All other material parameters and work conditions remained consistent.

Dynamic responses in different layers are illustrated in Table 12. Variation rules of the dynamic response amplitude and depth on subgrades filled using different materials are depicted in Figure 13.

**Table 12.** Dynamic response in different layers.

| Test Layer | Dynamic Acceleration (m·s$^{-2}$) | | | | | Dynamic Velocity (mm·s$^{-1}$) | | | | | Dynamic Displacement (×10$^{-3}$ mm) | | | | | Dynamic Stress (kPa) | | | | |
|---|---|---|---|---|---|---|---|---|---|---|---|---|---|---|---|---|---|---|---|---|
| | I | II | III | IV | V | I | II | III | IV | V | I | II | III | IV | V | I | II | III | IV | V |
| 1 | 16.5 | 14.5 | 14 | 13.5 | 12 | 9.4 | 7.8 | 5.8 | 5.4 | 5.1 | −82 | −63 | −57 | −56 | −53 | 60.5 | 63 | 65 | 68 | 73 |
| 2 | 7 | 7.5 | 8.2 | 7.8 | 7 | 7.7 | 5.4 | 4.4 | 4.1 | 3.8 | −60 | −52 | −50 | −48 | −48 | 17.5 | 29 | 30 | 32 | 35 |
| 3 | 4.8 | 4.8 | 4 | 4.8 | 4.6 | 6.5 | 4 | 3.1 | 2.7 | 2.6 | −50 | −40 | −38 | −38 | −36 | 11.8 | 17 | 20 | 22 | 23 |
| 4 | 2.3 | 2.2 | 2.2 | 2.2 | 2.1 | 3.3 | 3 | 2.1 | 1.8 | 3.3 | −25 | −25 | −24 | −25 | −22 | 4.0 | 6.4 | 6.5 | 7 | 7.3 |
| 5 | 2 | 1 | 0.8 | 1 | 0.9 | 1.1 | 1 | 0.9 | 0.8 | 1.1 | −10 | −10 | −10 | −10 | −10 | 2.4 | 3.5 | 3.5 | 3.6 | 3.6 |

Note: "I" is WAS +6% quicklime, "II" is WAS +3% cement, "III" is WAS +5% cement, "IV" is WAS +7% cement. "1" is the surface of bed (depth of −0.65259 m from rail surface), "2" is the bottom of surface bed (depth of −0.95259 m from rail surface), "3" is the bottom of bed (depth of −2.45259 m from rail surface), "3" and "4" are the subgrade layer (depth of −7.65259 m and −16.65259 m from rail surface).

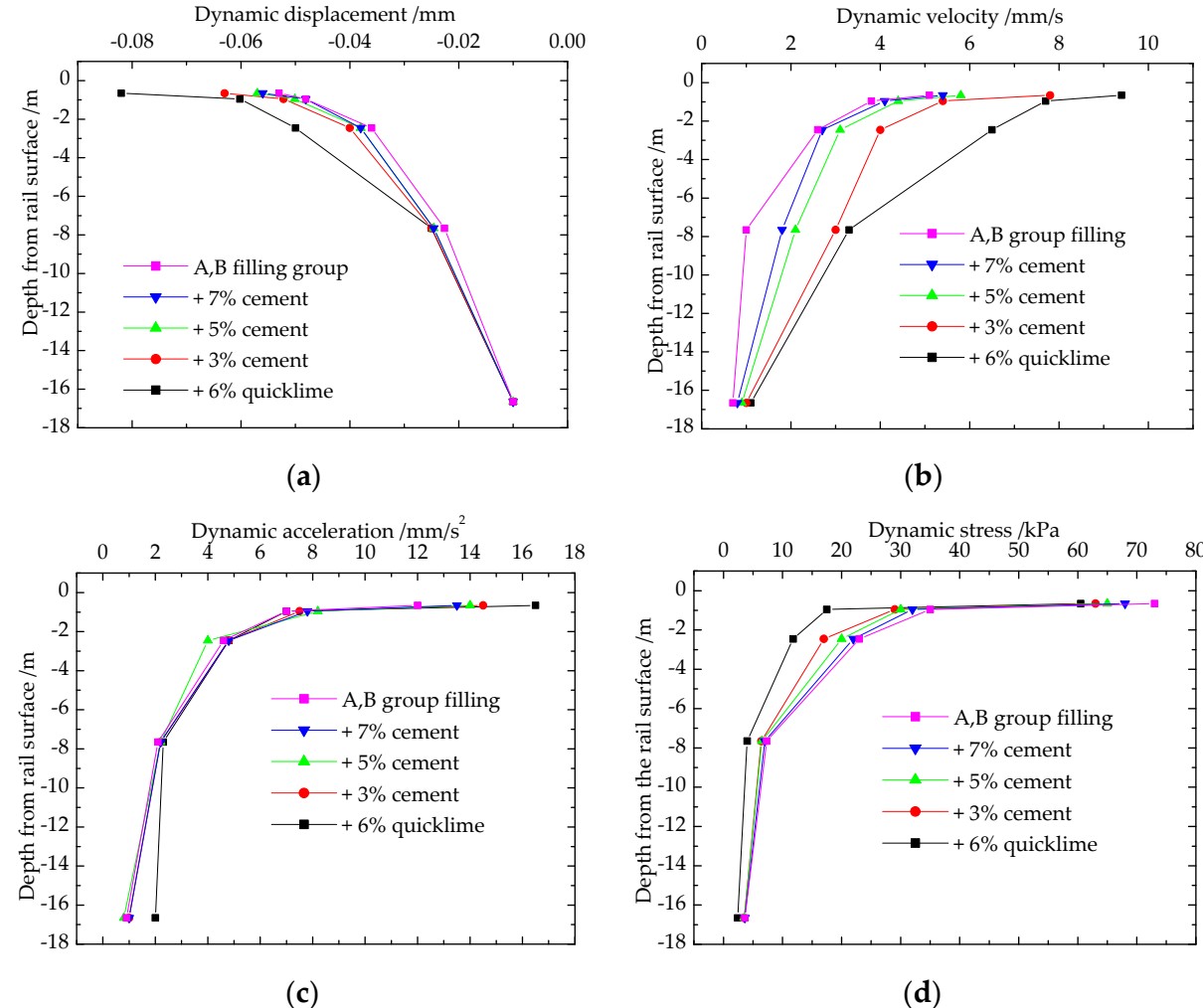

**Figure 13.** Dynamic responses at various depths for different fillings. (**a**) Displacement; (**b**) Velocity; (**c**) Acceleration; (**d**) Stress.

From Figure 13, The vertical displacement, velocity and acceleration on the bed surface were found to increase by 13.8%, 20.5% and 30.1%, respectively, from 5% quicklime + WAS to 3% cement + WAS; and by 3.5%, 3.3% and 10.5%, respectively, from 3% cement + WAS to 5% cement + WAS; and by 3.7%, 7.4% and 1.8%, respectively, from 5% cement + WAS to 7% cement + WAS; and by 12.5%, 5.8% and 5.6%, respectively, from 7% cement + WAS to A/B-group filling. The variation rules of vertical stress are the opposite. The results illustrate that the change rate of dynamic responses was much greater when the subgrade filling changed from quicklime-modified filling to A/B-group filling, than when the subgrade filling changed from cement-modified filling to A/B-group filling. It is inferred that the cement-modified sample is more appropriate as a substitute to A/B-group filling.

Additionally, the results indicate that vertical dynamic responses were attenuated rapidly after spreading through the subgrade, particularly on the bed surface. This may be attributed to the damping effect of the subgrade layer. The damping effect caused the train load to spread to deep layers by way of dynamic waves. The greater the stiffness, the faster the dynamic waves diffuse and attenuate. This indicates that with greater subgrade stiffness there is a deeper impact on dynamic influence. For a softer subgrade, greater dynamic response occurs at a shallow depth.

In addition, Figure 13a indicates that elastic deformation of the subgrade mainly occurred in the subgrade, particularly in the bed-surface layer. Moreover, the maximum dynamic displacement filled with the five materials was entirely within the deformation control allowance value, which is 1.5 mm in Japan, 2.0 mm in China and 1.0 mm in Europe.

### 4.2.2. Dynamics Strength Analysis

Dynamic strength was tested using a dynamic triaxial instrument, DTC-306, made in Japan, which was calibrated before testing. The sample was sieved through a 2 mm sieve and then was compacted to a $\Phi 35$ mm $\times$ 70 mm cylinder specimen with OMC at a compactness of 0.95. The test was conducted at confining pressures of 0 kPa, 25 kPa, 50 kPa and 100 kPa (stress-controlling) and a sine-wave load with 5 Hz frequency.

The dynamic strength of the specimen refers to the maximum stress at which the corresponding strain does not exceed a certain allowable value (1% in this study) under the given loading times. The dynamic strength curve, known as the fatigue curve, is usually described by the following power function:

$$\sigma_{d,f} = kN^{\theta} \tag{7}$$

where $k$ and $\theta$ are the test parameters, which are influenced by compactness, confining pressure, water content and load frequency; $\sigma_{d,f}$ is the maximum dynamic stress amplitude when the strain of specimen is 1%.

$10^8$ is converted from the number of vehicle axles passing through the Wuhan–Guangzhou HSR each year. It is taken as the loading times that a subgrade filling can bear in its service life. Table 13 illustrates the dynamic strength obtained by the fitting equation. Figure 14 shows the relationship between dynamic strength and loading times.

**Table 13.** Dynamic of modified WAS by fitting power function.

| Group | Natural Sample | +Quicklime | | | +Cement | | |
|---|---|---|---|---|---|---|---|
| $\chi$/% | 0 | 5 | 6 | 7 | 3 | 5 | 7 |
| k | 203.97 | 215.62 | 235.62 | 294.62 | 597.82 | 661.03 | 490.04 |
| $\theta$ | −0.0498 | −0.0301 | −0.0234 | −0.0381 | −0.0237 | −0.02658 | −0.00475 |
| $\sigma_{d,f}$/kPa | 66.17 | 123.84 | 153.10 | 146.03 | 386.30 | 417.00 | 448.99 |
| R | 0.9601 | 0.9623 | 0.9247 | 0.9748 | 0.9573 | 0.9488 | 0.9378 |

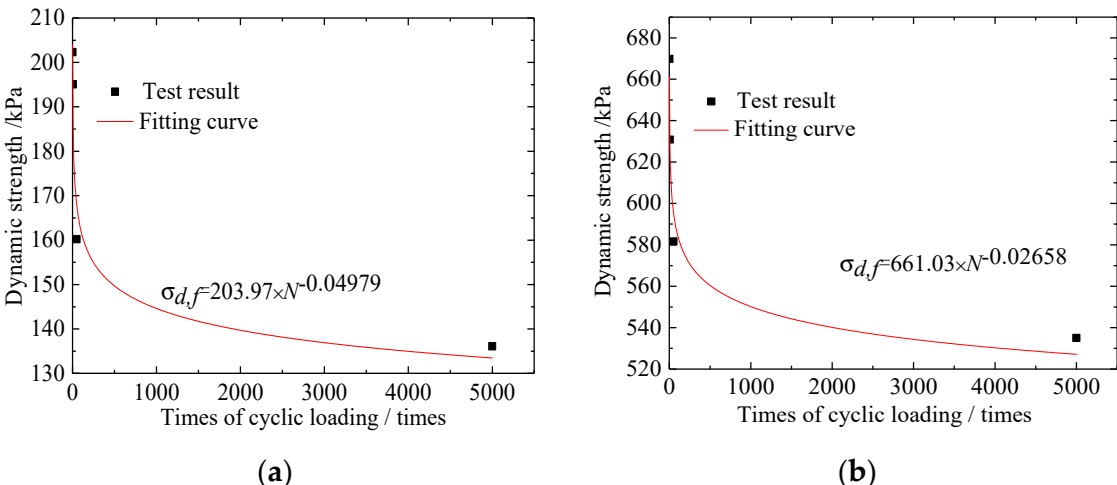

**Figure 14.** Variation of dynamic strength with cyclic loading times. (**a**) Natural sample; (**b**) +5% cement.

From Figure 13d, it is inferred that the dynamic stress on the bed layer is 73 kPa, and 23 kPa on subgrade layer. According to Formula (1), for the bed layer, $K_h$= 0.95, $\sigma_{zl} \leq 73$ kPa, $\sigma_{bcu} \geq 117.8/\eta_g$. If $\eta_g = 0.95$, $\sigma_{bcu} \geq 194$ kPa; if $\eta_g = 0.7$, $\sigma_{bcu} \geq 264.2$ kPa. Thus, the dynamic strengths of WAS with 3%, 5% and 7% cement meet the dynamic requirement, while the dynamic strength of WAS with quicklime cannot meet the dynamic requirement.

For the subgrade layer, $K_h = 0.90$, $\sigma_{zl} \leq 23$ kPa, based on Formula (1), $\sigma_{bcu} \geq 52.8/\eta_g$. If $\eta_g = 0.95$, $\sigma_{bcu} \geq 58.1$ kPa; if $\eta_g = 0.7$, $\sigma_{bcu} \geq 78.8$ kPa. Thus, the dynamic strengths of WAS with cement and quicklime can both meet the dynamic requirement.

*4.3. Disccusions*

The elastic deformation of the subgrade mainly occurred in the subgrade, particularly in the bed-surface layer. Moreover, the maximum dynamic displacement filled with the five materials (+5%, 6%, 7% quicklime and +3%, 5%, 7% cement) were all less than 1.0 mm, which is within the deformation control allowance value in Japan, China and Europe. In addition, the UCS of quicklime/cement mixtures were greater than 0.5 MPa when the compactness was 0.92, meeting the static strength requirements.

The dynamic performances of WAS were modified by mixing cement or quicklime. The variation rules of dynamic response changes along depth with cement-modified WAS are consistent with that of A/B-group filing, and the dynamic strength of cement-modified soil increased five- or six-fold, meeting the dynamic requirement, while the dynamic strength of quicklime-modified soil increased two-fold, failing to meet the dynamic requirement.

**5. Conclusions**

Adoption of lime and cement for the improvement of poor-subgrade soils has technical, economic and environmental advantages. This paper covers a comparative study between untreated samples and samples treated with different contents of lime and cement at different proportions. The results suggest that mixing quicklime/cement makes the content of muscovite decrease considerably, whereas it makes the content of kaolinite and clastic minerals—especially amorphous phase which is composed of $SiO_2$—increase greatly, and coarsens particles of WAS. Therefore, the physical, mechanical, and dynamic performances of the modified WAS are improved by adding quicklime or cement. However, the water stability of WAS was improved by adding cement, while mixing the WAS with quicklime showed weak water stability. Therefore, 5% cement is suggested to modify the WAS in order fill subgrades in HSR.

**Author Contributions:** Conceptualization, P.H. and W.G.; methodology, W.G.; software, P.H.; validation, P.H., W.G. and L.W.; formal analysis, P.H.; investigation, W.G.; resources, L.W.; data curation, P.H.; writing—original draft preparation, P.H.; writing—review and editing, W.G.; supervision, L.W.; project administration, W.G., L.W.; funding acquisition, P.H., W.G. All authors have read and agreed to the published version of the manuscript.

**Funding:** This work was supported by the National Natural Science Foundation of China (Funder:Wei Guo; Project No. 51878674, No. 52022113), Excellent Youth Fund of Hunan Provincial Department of Education (Funder: Ping Hu; No.20B098), Natural Science Foundation of Hunan Province (Funder: Ping Hu; No. 2020JJ5488). These supports are gratefully acknowledged.

**Conflicts of Interest:** The authors declare no conflict of interest.

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
