# Peer review of "Applicability Study on Modified Argillaceous Slate as Subgrade Filling for High-Speed Railway"

_applsci, doi:10.3390/app12042227_

Round 1

Reviewer 1 Report

The paper presents the results of experiments and numerical analyzes of argillaceous slate modification as track bed filling for high-speed railways in the form of: cement and quicklime additives.

The work is interesting, but quite chaotic and not in line with the MDPI standard. The work contains many editorial errors, e.g.:

  • line 10, 11, 31, ... - missing spaces before abbreviations
  • line 33, 39, 42, 43, 44, 46, 55, ... - missing spaces before refs in []
  • line 72 - wrong formatting of new section
  • line 73 - wrong formatting of subsection
  • line 120-126  - wrong font (smaller)
  • line 130-134 - wrong font size
  • figure 2 - to small figure - font used in fig should be similar as in the main text
  •  line 230 - fig7 instead of Figure 7
  • and many more, mainly wrong format, missing spaces and text repetition.

The introduction is short and general, a broader view of the subject is, in my opinion, necessary.

Chapter 3 should be divided into two - 3) Results - with a short presentation of the results and 4) Discussion - with a longer discussion of the results obtained.

Chapters 4 and 5 should be integrated into new chapters 3-Results and 4_Discussion as subsections.

Conclusions in the form of sub-points are not a good habit. The last chapter should contain a short summary of the research carried out.

Reviewer 2 Report

Reviewer comments as attached

Round 2

Reviewer 1 Report

The manuscript has been revised according to the reviewers' recommendations, although the reordering of subsections is not in line with my suggestions. However, I believe it can be accepted for publication